# Consumption and Sources of Saturated Fatty Acids According to the 2019 Canada Food Guide: Data from the 2015 Canadian Community Health Survey

**DOI:** 10.3390/nu11091964

**Published:** 2019-08-21

**Authors:** Stéphanie Harrison, Didier Brassard, Simone Lemieux, Benoît Lamarche

**Affiliations:** 1Centre Nutrition, santé et société (NUTRISS), Institute of Nutrition and Functional Foods (INAF), Université Laval, Québec, QC G1V 0A6, Canada; 2School of Nutrition, Université Laval, Québec, QC G1V 0A6, Canada

**Keywords:** Canada, Canadian adults, Canada’s Food Guide, CCHS 2015, saturated fats

## Abstract

The 2019 revised version of Canada’s Food Guide (CFG) recommends limiting the consumption of processed foods that are high in saturated fatty acids (SFA). Yet, the contributions of each CFG group to the total SFA intake of Canadians are not specifically known. The objectives of this study were to quantify the total SFA intake of Canadians, determine the sources of SFA consumed by Canadian adults, and identify potential differences in these sources. A nation representative sample from the Canadian Community Health Survey (CCHS – Nutrition 2015) was used for these analyses. Dietary intakes were measured using a single 24-h recall. Food sources of SFA were classified according to the revised 2019 CFG categories. We have also examined the contribution of foods not included in these three categories to total SFA intake. Among Canadian adults, total SFA contributed to 10.4 ± 0.1% (SE) of total energy intake (E). The “Protein foods” (47.7 ± 0.5% with 23.2 ± 0.4% from *milk and alternatives* and 24.5 ± 0.4% from *meats and alternatives*) and “All other foods” (44.2 ± 0.5%) categories were the main sources of total SFA intake. Few differences in SFA sources were identified between sexes, age groups, education levels, and body mass index (BMI) categories. These data show that the mean SFA consumption is greater than the 10% E cut-off previously proposed in Canada. Future studies should examine which food substitution is most likely to contribute to a greater reduction in SFA intake at the population level.

## 1. Introduction

Dietary saturated fatty acids (SFA) increase low-density lipoprotein (LDL) cholesterol (LDL-C) when compared with carbohydrates and polyunsaturated fats [1]. Accordingly, the replacement of SFA by polyunsaturated fats reduces the risk of cardiovascular diseases (CVD) [1,2]. However, many studies emphasize that the association between SFA and CVD risk may depend on its dietary source [3,4,5]. For example, the intake of SFA from dairy products such as cheese and milk showed no association with the risk of CVD, while the intake of SFA from meats was positively associated with the risk of CVD [4,5]. Nonetheless, the 2019 revised version of Canada’s Food Guide (CFG) recommends limited intakes of all foods high in SFA, sodium, and sugars [6]. Additionally, Canada’s dietary guidelines recommend that foods containing mostly unsaturated fatty acids replace foods that contain mostly SFA, irrespective of the food source [6]. Since SFA is a nutrient of public health interest, it is important to quantify SFA intake in Canada and identify the predominant food sources of SFA within the new CFG 2019 categories in the Canadian diet, which are currently unknown.

Therefore, the objectives of this study were to quantify the SFA intakes of Canadian adults, determine the sources of SFA consumed by Canadian adults, and identify how these vary according to various sociodemographic characteristics and body weight status. We hypothesized that the mean SFA intake of Canadian adults is above current recommendations, and that foods not included in any of the three new CFG categories are the most important contributors to SFA consumption in Canadian adults. We also hypothesized that men consume more SFA from protein foods than women, that body weight is positively associated with total SFA intake, and that age is inversely associated with the proportion of SFA from foods not included in the CFG categories.

## 2. Materials and Methods

### 2.1. Study Population

Data from the nutrition component of the 2015 Canadian Community Health Survey (CCHS Nutrition) was used for this study. The CCHS Nutrition is a sample survey with a cross-sectional design providing information on the eating habits and dietary intakes of Canadians aged 1 year old and over living in the 10 Canadian provinces. People living on reserves and other Aboriginal settlements, full-time members of the Canadian Forces, and institutionalized individuals were excluded from the survey’s coverage. The sampling method, based on age, sex, geography, and socioeconomic status was designed in order to generate a sample representative of the Canadian population. For the present analyses, only Canadian adults were included (i.e., individuals of <19 years or >70 years were excluded). Weight status was qualified according to body mass index (BMI). The present study objectives were pre-specified in contract #17-SSH-LAVAL-5405 with Statistics Canada.

### 2.2. Dietary Data Collection

Dietary data were obtained using a single computer-assisted 24-h recall administered by trained interviewers using the Automated Multiple-Pass Method (AMPM) [7]. The AMPM consists of five steps: a quick list of foods easily remembered by the respondent, a list of nine categories of foods often forgotten by respondents (e.g., beverages, sweets, and snacks), time and occasion of consumption of foods consumed at the same time, details on each food consumed and where it was consumed, and a final probe to make sure no foods were forgotten [7]. Energy and nutrient intakes were derived from the Canadian Nutrient File (CNF—2015 version). The CNF was representative of foods available on the market at the time of the survey [8].

SFA food sources were classified according to the new 2019 CFG categories: (1) vegetables and fruits (excluding fruit and/or vegetable juices), (2) whole grain foods (excluding refined grains), and (3) protein foods, which include the previous CFG categories *meats and alternatives* and *milk and alternatives*. The contribution to total SFA intake of “All other foods”, i.e., all foods not included in the three new food categories of the CFG such as fruit juices, refined grains, and salty snacks, was also examined.

### 2.3. Statistical Analyses

All the analyses were undertaken using survey-specific procedures to account for the CCHS Nutrition design and balanced repeated replication (BRR) for variance estimation. The sampling weights and 500 bootstrap weights provided by Statistics Canada were used to achieve the representativeness of the study sample. Thus, the results presented here are representative of the entire adult Canadian population aged 19 to 70 years with exclusion of the aforementioned groups. Means, medians, and variances of continuous variables were obtained using PROC SURVEYMEANS. Proportions/variances of categorical variables were computed using PROC SURVEYFREQ. Potential differences among sociodemographic characteristics and weight statuses were assessed using general linear models in PROC SURVEYREG with Tukey–Kramer adjustment for multiple comparisons when appropriate. SAS (v. 9.4) was used for all the analyses. *p*-values lower than 0.05 were considered statistically significant.

## 3. Results

### 3.1. CCHS Sample

The mean age (±SE) of the sample used for analyses was 45.0 ± 0.3 years, with 50.2% of respondents being female and 48.3% having a household income greater than $80,000/year. The sample was evenly distributed among education levels, with 33.9% having a high school diploma or no diploma, 31.0% having a CEGEP, college, or trade school certificate, and 34.4% with a University degree. CEGEP is a pre-university and technical college institution that is specific to the Province of Quebec. Over one-third (39.3%) of Canadian adults were in the normal weight category (BMI <25 kg/m^2^), 34.1% were overweight (BMI 25–30 kg/m^2^), and 26.6% were obese (BMI >30 kg/m^2^).

### 3.2. Saturated Fatty Acids Consumption

SFA contributed to 10.4% of total energy intake (E) among Canadian adults in 2015 (Table 1). No difference was found between sexes and age groups. Compared with normal-weight individuals, adults with obesity consumed slightly more SFA (+0.6%E, *p* = 0.05). Canadians with a CEGEP, college, or trade certificate consumed more SFA than Canadians with a high school diploma or no diploma, or with a University degree (10.8% E versus 10.2% E and 10.2% E, respectively; *p* = 0.005 and *p* = 0.003, respectively). There were only small differences in total SFA consumption among the 10 Canadian provinces (Table 1).

Figure 1 shows the relative contribution of each 2019 CFG category to total SFA intake. We also examined the relative contribution of the former CFG categories, *milk and alternatives* and *meats and alternatives*, to total SFA intake. “Protein foods” (47.8% of total SFA intake) and “All other foods” (44.2% of total SFA intake) were the main sources of SFA in the diet of Canadian adults in 2015. *Milk and alternatives* and *meats and alternatives* equally contributed to SFA from the “Protein foods” category. Whole grain foods were the least important contributor to total SFA intake (2.5% of total SFA intake).

Table 2 shows the relative contribution of each 2019 CFG category to total SFA intake according to sex, age, BMI, and education level. Men consumed more SFA from the “Protein foods” category (49.5% of total SFA intake) and less from the “All other foods” category (42.8% of total SFA intake) than women (46.0% and 45.7% of total SFA intake, respectively). Additionally, men consumed more SFA from *meats and alternatives* than women (+4.3% of total SFA intake, *p* < 0.001). No differences were found between age groups for the “Protein foods” category. However, younger adults (19–30 years old) consumed more SFA from *milk and alternatives* and less from *meats and alternatives* than older adults (31–50 years old and 51–70 years old). Inversely, older adults (51–70 years old) consumed more SFA from whole grain foods than younger adults (19–30 years old). We also found that Canadians with a CEGEP, college, or trade certificate consumed less SFA from whole grain foods than Canadians with a high school diploma or no diploma, or Canadians with a University degree (Table 2). Finally, no difference in SFA sources was identified among BMI categories. Figure 2 presents the proportion of total SFA intake from the “Protein foods” and “All other foods” categories. Respondents living in the province of Quebec consumed more SFA from the “Protein foods” category than respondents living in Newfoundland and Labrador and Manitoba (+6.1% of total SFA intake, both *p* < 0.05). Respondents living in Newfoundland and Labrador and in Prince Edward Island had a higher proportion of their SFA coming from the “All other foods” category than Canadians from Quebec, Ontario, and British Columbia (Figure 2). 

## 4. Discussion

Increased consumption of SFA is known to raise LDL-C, which is an established risk factor of CVD, making it a target nutrient for public health policies [1]. However, data suggest that the association between SFA consumption and CVD risk may vary according to the dietary source of SFA [4,5]. Therefore, it is important from a public health perspective to identify the sources of SFA consumed at a population level to best inform future public health initiatives on healthy eating. In the present study, we have assessed the proportion of total energy intake consumed as SFA by Canadian adults as well as the main sources of SFA in the Canadian diet, based, for the first time, on the revised CFG categories. We found that in 2015, Canadian adults consumed on average slightly more than 10% E as SFA. Moreover, “Protein foods” and “All other foods” were the main categories contributing to total SFA consumption in adults from Canada. Few differences in SFA sources were identified between sexes, age groups, education levels, and weight statuses.

### 4.1. Saturated Fatty Acids Consumption in Canadian Adults

The World Health Organization (WHO) currently recommends limiting the intake of SFA at a maximum of 10% E per day [9]. Here, we found that in 2015, Canadian adults consumed 10.4% E as SFA, which is slightly higher on average than this recommendation. In Canada, Gray-Donald et al. (2000) documented, using data from the Food Habits of Canadians survey (1997–1998), that the mean SFA intake of adults varied between 9.5–10.2% E, according to sex and age [10]. Health Canada reported a mean SFA intake between 10.0–10.5% E among Canadian adults, depending on sex and age, based on consumption data collected in 2004 as part of cycle 2.2 of the CCHS [11]. This suggests that despite years of public health initiatives aimed at improving diet quality and reducing SFA intake in Canada, SFA intake has apparently remained stable over the last 20 years.

Multiple studies have observed intakes higher than the WHO recommendation at a population level. For example, Huth et al. (2013) found that Americans aged 2 years or older consume a mean of 11.4% E as SFA [12]. Among European countries, Eilander et al. (2015) found that SFA intake varied between 8.9–15.5%E, most countries having a mean consumption above recommendations [13]. Nevertheless, in 2010, the global mean consumption of SFA represented 9.4% E, suggesting that some countries successfully reached intakes below the recommended threshold [14]. It is important to stress that most studies cited above included children, adults, and the elderly, while only adults were considered here.

Data from the present study suggest a weak association between education level and SFA intake (*p* = 0.002); respondents with a CEGEP, college, or trade certificate consumed more SFA than respondents with a lower education (high school diploma or no diploma) or a higher education (University degree). This “non-linear” association between SFA intake and education level is unclear. Also, very few studies have examined the potential associations between SFA consumption and education level, making comparison difficult. However, Hiza et al. (2012) found that American adults with no diploma had higher Healthy Eating Index (HEI) subscores for SFA than those with a higher education level, reflecting a lower intake [15]. Therefore, more studies are needed to better understand how education relates to SFA intake specifically, and whether it is important from a public health perspective or not.

No difference in total SFA intake, which is reported relative to total energy intake, was identified between sexes and age groups in the present study, which is consistent with the available literature on the subject [14,15,16]. Total SFA intake was also similar across all 10 Canadian provinces. Finally, data suggest that adults with obesity consume slightly more SFA than adults with a normal BMI (+0.6% E, *p* = 0.05), which is consistent with previous data in the American adult population and in the Canadian population [17,18]. However, Raatz et al. analyzed total SFA intake in grams per day, while we used percentage of total energy intake (% E). When looking at total SFA consumption in grams per day in Canadian adults, intakes across all the weight statuses were similar (not shown). We cannot exclude the possibility that other lifestyle-related factors, such as physical activity, smoking, and stress, among others, have influenced this association between SFA intake and obesity. However, it is important to stress that even if some differences in total SFA intake were identified between the studied sociodemographic characteristics, the mean intake of SFA was above the recommended 10% E for all the studied subgroups.

### 4.2. Contribution of 2019 CFG Food Categories to Total Saturated Fatty Acids Intake

As expected, “Protein foods” (47.8%), and “All other foods” not included in the 2019 CFG (44.2%) were the two main sources of SFA in the diet of Canadian adults in 2015. To our knowledge, this is the first study to evaluate the contribution of each 2019 CFG category to total SFA intake, since most studies available to date have reported data using more specific dietary sources of SFA such as milk, cheese, or red meat. While comparison with data from previous studies is limited by our use of this novel classification of food groups, data are nevertheless consistent with results from studies available on the subject [12,13,19]. For example, Huth et al. found that cheese was the main SFA source in the American diet, followed by beef, milk, and fats and oils other than margarine or butter [12]. In the United Kingdom, fats and oils (19%) were the main source of SFA consumed in the household, followed by meats and meat products (15%), milk and cream (14%), and cheese (10%) [19].

Very few studies have documented the associations between food sources of SFA and sociodemographic characteristics. Kirkpatrick et al. (2019) have shown, based on data from the CCHS Nutrition 2015, that the main singular food sources of SFA in the Canadian diet were red meat mixed dishes, unflavored milk, cheese, egg dishes, and dairy-based desserts [20]. They reported that the top food sources of SFA in Canada were similar across all income groups [20]. They also found that milk was among the top five sources of SFA in all age–sex groups in Canadians aged 2 years of more [20]. Results were not reported according to education levels and BMI categories or according to the new food categories of the 2019 CFG. Additionally, while Kirkpatrick et al. looked at age–sex groups, we looked at age and sex separately, and found that relatively speaking, men consume more SFA from protein foods than women, mostly due to a higher proportion of SFA from the *meats and alternatives* category (+4.3% of total SFA intake versus women, *p* < 0.001).

When looking at potential differences in SFA sources between age groups, we found that younger adults (19–30 years old) consumed less SFA from *meats and alternatives* and more from *milk and alternatives* than older adults (31–50 and 51–70 years old, all *p* < 0.05). This can be due to either a lower consumption of meat or consumption of meat with a lower fat content among younger adults in Canada. A lower consumption of meat among younger adults is unlikely, based on findings by Tugault-Lafleur and Black (2019) [21]. Indeed, Tugault-Lafleur and Black found that the consumption of *meats and alternatives* has increased by 0.3 servings/day in Canadian adults 18–54 years old between 2004–2015 [21]. However, they also reported that the consumption of meat and poultry, fish and shellfish, and processed meats did not change between 2004–2015 in adults 18–54 years old, suggesting that the increase in the consumption of *meats and alternatives* is not due to the consumption of meat *per se* [21]. Additionally, these data were based only on adults aged between 18–54 years. Therefore, the extent to which meat consumption has changed between 2004–2015 among older adults in Canada is unknown. Nevertheless, Daniel et al. have shown, based on NHANES 2003–2004 data, that meat intake declines with age in American adults [22].

Finally, the proportion of SFA from each 2019 CFG category was similar across all BMI categories. This result was unexpected, considering that obese individuals generally report consuming more foods that have a low nutritive value than non-obese individuals [23]. It is possible that the absence of difference in food sources of SFA among BMI categories is due, at least partly, to the well-documented under-reporting of lower nutritive value foods by obese individuals [24,25]. Data from the present study also showed geographical differences among provinces in the proportion of SFA coming from the “Protein foods” and “All other foods” categories. This suggests that public health initiatives aimed at lowering total SFA consumption may need to be developed according to regional differences and specificities.

### 4.3. Strengths and Limitations

To our knowledge, this is the first study to assess SFA dietary sources using the new 2019 CFG food categories. The assessment of potential differences in SFA intakes and dietary sources between sexes, age groups, education levels, and weight statuses is also a strength, as few studies are currently available on the subject. The use of survey-specific procedures applied to data from CCHS 2015 implies that results are representative of all Canadian adults, which is an important strength. Dietary data was obtained using a 24-h recall, which is affected by within-person variation, mostly caused by the day-to-day variation of food intakes [26]. As a result, a single 24-h recall is not reflective of a given respondent’s usual intakes. Regardless, a single 24-h recall can be reflective of the usual intakes at a population level, which made it suitable for the current analyses [26]. Furthermore, there is documented under-reporting in CCHS 2015, suggesting that the true intake of SFA among Canadians may be higher than what was found here [27].

## 5. Conclusions

These findings suggest that dietary intake of SFA among Canadian adults in 2015 was in average above the 10% E recommendation. Similar results from previous surveys in Canada also suggest that SFA intake has not changed significantly over the past years, despite public health efforts aimed at improving the diet quality of Canadians. Unsurprisingly, almost half of SFA consumed by Canadians in 2015 came from foods that are not recommended and/or not included in the CFG. Therefore, even if the main sources of total SFA intake are protein foods, future health policies should focus primarily on reducing the population’s intake of processed foods and/or foods not included in the CFG, as they are an important source of SFA. Such focus on low nutritive quality foods are entirely in line with Canada’s most recent dietary guidelines, in which a reduction in the intake of processed foods high in sodium, sugar, and SFA is strongly recommended. Future studies should examine which food substitutions are likely to contribute the most to the reduction of SFA consumption in Canada.

## Figures and Tables

**Figure 1 nutrients-11-01964-f001:**
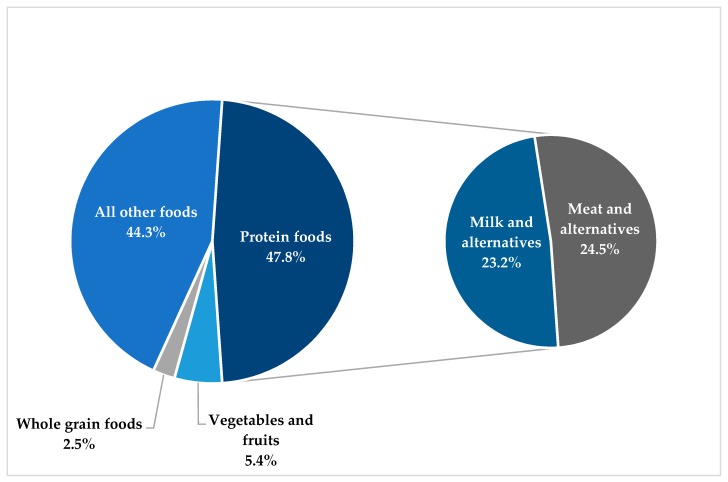
Contribution of 2019 Canada’s Food Guide categories to total SFA intake (in % of total SFA intake).

**Figure 2 nutrients-11-01964-f002:**
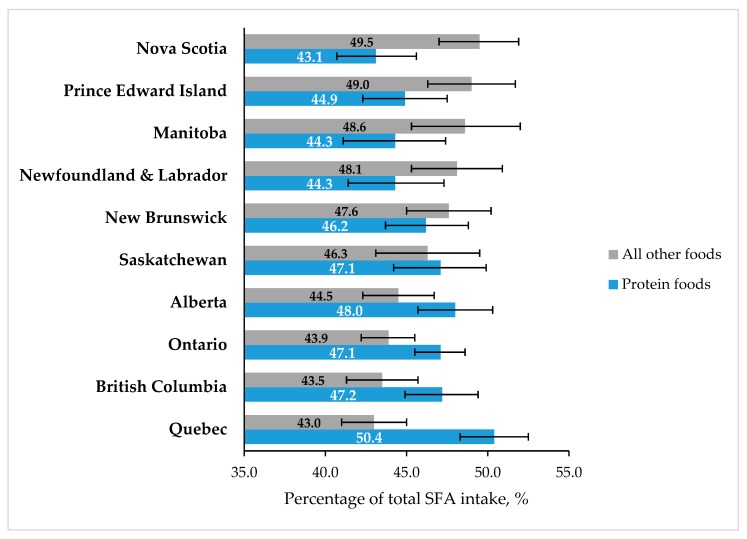
Proportion of total SFA intake from the “All other foods” and “Protein foods” categories, according to province of residence.

**Table 1 nutrients-11-01964-t001:** Saturated fatty acids (SFA) consumption in all Canadian adults and according to sex, age group, education level, body mass index, and province of residence.

	% of Total Energy Intake ^‡^	95% CI		*p*-value *
Canadian adults	10.4	10.2	10.5	–
Sex				0.49
Women	10.4	10.2	10.6	
Men	10.3	10.1	10.5	
Age				0.05
19–30 years	10.6	10.2	10.9	
31–50 years	10.5	10.2	10.7	
51–70 years	10.2	10.0	10.4	
Body mass index				0.07
Normal	10.3	10.0	10.5	
Overweight	10.4	10.1	10.7	
Obesity	10.8	10.4	11.2	
Education level				0.002
High School/No diploma	10.2 ^a^	10.0	10.4	
CEGEP ^†^/Trade certificate/college	10.8 ^b^	10.6	11.1	
University	10.2 ^a^	9.9	10.4	
Unknown	9.3 ^a,b^	7.6	11.1	
Province of residence				<0.0001
Newfoundland & Labrador	10.1	9.7	10.5	
Prince Edward Island	11.0	10.6	11.4	
Nova Scotia	10.9	10.5	11.3	
New Brunswick	10.8	10.2	11.3	
Quebec	10.6	10.2	11.0	
Ontario	10.2	10.0	10.5	
Manitoba	10.7	10.2	11.2	
Saskatchewan	10.5	9.9	11.1	
Alberta	10.6	10.2	11.0	
British Columbia	10.0	9.7	10.4	

* *p*-values were found using a linear regression model with Tukey–Kramer adjustment for multiple comparisons. CI: confidence intervals. ^†^ CEGEP is a pre-university and technical college institution. specific to the Province of Quebec. ^‡^ Subgroups without a common superscript letter are significantly different (*p* < 0.05, Tukey–Kramer).

**Table 2 nutrients-11-01964-t002:** Contribution of each 2019 Canada’s Food Guide categories to total SFA intake according to sex, age, body mass index (BMI), and education level in Canadian adults (19–70 years old).

	Vegetables and Fruits *	Whole Grain Foods	Protein Foods (All)	Milk and Alternatives	Meats and Alternatives	All Other Foods
Sex						
Women	5.6 (5.2–6.0)	2.7 (2.4–3.0)	46.0 (44.8–47.1) ^a^	23.6 (22.6–24.6)	22.4 (21.4–23.3) ^a^	45.7 (44.5–46.9) ^a^
Men	5.2 (4.8–5.6)	2.4 (2.1–2.6)	49.5 (48.0–50.9) ^b^	22.7 (21.6–23.9)	26.7 (25.5–28.0) ^b^	42.8 (41.4–44.1) ^b^
Age						
19–30 years	5.0 (4.4–5.7)	2.1 (1.7–2.6) ^a^	48.3 (46.1–50.4)	26.4 (24.3–28.6) ^a^	21.8 (20.1–23.6) ^a^	44.5 (42.3–46.7)
31–50 years	5.6 (5.1–6.1)	2.5 (2.1–2.8) ^a,b^	47.9 (46.5–49.4)	22.8 (21.6–24.0) ^b^	25.2 (23.9–26.4) ^b^	43.9 (42.4–45.4)
51–70 years	5.3 (4.9–5.7)	2.8 (2.4–3.2) ^b^	47.1 (45.9–48.4)	22.0 (20.9–23.1) ^b^	25.2 (24.0–26.3) ^b^	44.4 (43.2–45.7)
BMI						
Normal (<25 kg/m^2^)	5.1 (4.5–5.7)	2.8 (2.3–3.2)	46.3 (44.4–48.3)	27.7 (21.1–24.4)	23.6 (22.1–25.0)	45.7 (43.8–47.7)
Overweight (25–30 kg/m^2^)	5.2 (4.7–5.7)	2.5 (2.0–3.0)	48.7 (46.9–50.5)	23.9 (22.4–25.4)	24.8 (22.9–26.7)	43.6 (41.9–45.3)
Obesity (≥30 kg/m^2^)	5.8 (5.1–6.5)	2.2 (1.7–2.6)	46.9 (44.7–49.0)	22.2 (20.5–23.9)	24.7 (23.0–26.4)	45.1 (42.9–47.3)
Education level						
High School/No diploma	4.8 (4.4–5.3)	2.6 (2.2–2.9) ^a^	47.0 (45.6–48.4)	22.5 (21.2–23.7)	24.5 (23.2–25.9)	45.5 (44.0–47.0)
CEGEP ^†^/Trade certificate/College	5.6 (5.0–6.1)	1.8 (1.6–2.1) ^b^	48.7 (47.0–50.4)	23.6 (22.2–25.1)	25.1 (23.8–26.4)	43.8 (42.1–45.5)
University	5.7 (5.2–6.3)	3.2 (2.7–3.6) ^a^	47.6 (46.0–49.2)	23.6 (22.2–24.9)	24.1 (22.6–25.5)	43.4 (41.8–44.9)
Unknown	7.2 (3.7–1.1)	1.4 (0.7–2.1) ^b,c^	44.0 (36.3–51.8)	18.2 (11.1–25.3)	25.8 (19.7–31.9)	47.4 (38.7–56.1)

Values are presented as mean percentage of total SFA intake (95% CI). * Subgroups without a common superscript letter are significantly different (*p* < 0.05, Tukey–Kramer). ^†^ CEGEP is a pre-university and technical college institution specific to the Province of Quebec.

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
