# Peer review of "Consumption and Sources of Saturated Fatty Acids According to the 2019 Canada Food Guide: Data from the 2015 Canadian Community Health Survey"

_nutrients, 2019, doi:10.3390/nu11091964_

Round 1

Reviewer 1 Report

Dear Authors

this study is unique and is highly needed in order to identify the level and sources of SFA intake, which could support the implementation of the FBDG in Canada; however I feel more analysis is needed regarding the :

1- The most popular foods high in SFA consumed  by Canadian which includes high levels of SFA .

2- the level of SFA consumption by Geographical areas.

If data are available, you may add one table reflecting the above points.

Best regards

Author Response

Reviewer 1

Point 1 :

This study is unique and is highly needed in order to identify the level and sources of SFA intake, which could support the implementation of the FBDG in Canada; however I feel more analysis is needed regarding the :

The most popular foods high in SFA consumed  by Canadian which includes high levels of SFA . the level of SFA consumption by Geographical areas.

If data are available, you may add one table reflecting the above points.

Response 1:

Thank you for these suggestions. As mentioned in our paper, Kirkpatrick et al(2019) published the main food sources of SFA consumed in Canada based on the CCHS 2015 data. Therefore, we believe that including such information to the current paper is redundant. However, we now more specifically refer to the data presented by Kirkpatrick et al and provide more information on the most popular/important sources of SFA in CCHS 2015 (lines 219-221).

On the other hand, we have included data on the total amounts of SFA consumed (Table 1) as well as proportions of SFA intake from the protein foods and «all other foods» categories (Figure 2) by geographical areas, i.e. for all Canadian provinces. The Results and Discussion sections were revised accordingly (Lines 107-108, 138-144, 197 and 248-251)

Reviewer 2 Report

The study concentrates on the saturated fatty acid intake in Canada from different sources as a parameter which would directly influence human health, as it is well known that a high consumption of SFA would increase the likelihood of developing a myriad of diseases.  This would also serve the Canadian health authorities to suggest amounts of SFA to be consumed and to update the different Guidelines for health in Canada, aiming at improving health and also to try to lower SFA intake.

In order to improve the manuscript, there are a few points I would like to mention for considerations by the authors.

1.- Although fish is considered the main source of unsaturated fatty acids (PUFAs, HUFAs) in human diet, they also contain SFA which would account for total SFA intake in a lower degree, but would contribute to better health and lower total SFA, as being also a protein source.  Three major SFA sources are mentioned in the text, I wonder if the authors collected data on fish consumption and could include that info in the text, and its contribution to total SFA.

I consider that the study its graphs and tables would be more interesting if the consumption of fish would be included, which contributes to a lesser extend in the SFA intake but would improve Canadian health and health guidelines.

Please respond or provide that information/data.

2.- Lines 104-106. Sentences a bit confused.  I do not know the educational system in Canada, as the authors also classified the volunteers according to their educational level splitting them in three groups (I would understand from lowest to hightest level): 1) CEGEP, college or trade certificate; 2) High school diploma or less; 3) University degree or higher.  Group 2 is confusing, particularly when reading sentence in line 106. High school diploma group is clear to me, but when added "or less", I interpret that "less" would mean that those individuals would then fall into CEGEP, college or trade certificate group (I guess by reading the order of the written list CEGEP would be a lower education than High school, but my common sense does not tell me that), mixing them up.

Is this the case? Wouldn't be better to removed "or less" in High school diploma group?  Or perhaps these could have been listed in a better order starting from the lowest education level which I understand is (1 lowest) High school diploma or less (it would be correct then to maintain this full name); (2 intermediate) CEGEP, college or trade certificate, and (3 highest) University of more. 

In any case, if I got it wrong and the name of group 2 is maintained, the current sentence needs a comma to better interpretation like "....compared to Canadians with a high school diploma or less, (comma) or with a University degree or more"

The same would apply to lines 96, 134, 172, 178.

3.- Lines 171-180.  It is stated in this study that education level was associated to SFA intake as energy source. Being lower for University degree and High School diploma or less, but higher for the CEGEP, college or trade certificate, but that there were not differences between the highest education=University and the lowest=High school diploma.

The association is difficult to see as the highest and lowest education level show no difference. Please, rephrase it as now the sentence is very affirmative but the data do not indicate a clear association with education level.

In order to see a real association one would expect to see the lowest SFA intake for Univ. group (highest educational level), followed by CEGEP group, and the highest SFA intake for the High school group (lowest educational level) but it is not the case.  Furthermore, Table 2 does not show that association either, just some differences for data on Whole grain food which doest sustain such statement.

Rectify it please.

Perhaps here other factors such as sedentarism, practice or not of healthy habits, sugar consumption, etc. my be affecting the data. Particularly in relation to just a 0,5% higher SFA intake for the obese individuals in the study. I would expect increases higher than 1-2% to be the cause to become obese.  Can the authors develop/discuss more about this little increase in SFA intake for the obese in relation to other possible factors.

Author Response

Reviewer 2

Point 1:

Although fish is considered the main source of unsaturated fatty acids (PUFAs, HUFAs) in human diet, they also contain SFA which would account for total SFA intake in a lower degree but would contribute to better health and lower total SFA, as being also a protein source.  Three major SFA sources are mentioned in the text, I wonder if the authors collected data on fish consumption and could include that info in the text, and its contribution to total SFA.

I consider that the study its graphs and tables would be more interesting if the consumption of fish would be included, which contributes to a lesser extend in the SFA intake but would improve Canadian health and health guidelines.

Please respond or provide that information/data.

Response 1:

Thank you for this comment. We agree that fish contains small amounts of SFA, while being a highly recommended food in dietary guidelines. As mentioned in our reply to the comments from the other reviewer, Kirkpatrick et al published a paper earlier this year concerning the main food sources of sodium, SFA and sugar in the Canadian diet based on data from CCHS Nutrition 2015. In their paper, any form of fish was not among the top 20 food sources of SFA in Canada. We therefore feel that focusing on this single food is beyond the scope of our objective, which was to quantify SFA consumption in Canada and to identify its main contributors according to the new 2019 Canada’s Food Guide categories (which doesn’t consider fish as a category).

Points 2 and 3:

Lines 104-106. Sentences a bit confused.  I do not know the educational system in Canada, as the authors also classified the volunteers according to their educational level splitting them in three groups (I would understand from lowest to hightest level): 1) CEGEP, college or trade certificate; 2) High school diploma or less; 3) University degree or higher.  Group 2 is confusing, particularly when reading sentence in line 106. High school diploma group is clear to me, but when added "or less", I interpret that "less" would mean that those individuals would then fall into CEGEP, college or trade certificate group (I guess by reading the order of the written list CEGEP would be a lower education than High school, but my common sense does not tell me that), mixing them up.

Is this the case? Wouldn't be better to removed "or less" in High school diploma group?  Or perhaps these could have been listed in a better order starting from the lowest education level which I understand is (1 lowest) High school diploma or less (it would be correct then to maintain this full name); (2 intermediate) CEGEP, college or trade certificate, and (3 highest) University of more. 

In any case, if I got it wrong and the name of group 2 is maintained, the current sentence needs a comma to better interpretation like "....compared to Canadians with a high school diploma or less, (comma) or with a University degree or more"

The same would apply to lines 96, 134, 172, 178.

Lines 171-180.  It is stated in this study that education level was associated to SFA intake as energy source. Being lower for University degree and High School diploma or less, but higher for the CEGEP, college or trade certificate, but that there were not differences between the highest education=University and the lowest=High school diploma.

The association is difficult to see as the highest and lowest education level show no difference. Please, rephrase it as now the sentence is very affirmative but the data do not indicate a clear association with education level.

In order to see a real association one would expect to see the lowest SFA intake for Univ. group (highest educational level), followed by CEGEP group, and the highest SFA intake for the High school group (lowest educational level) but it is not the case.  Furthermore, Table 2 does not show that association either, just some differences for data on Whole grain food which doest sustain such statement.

Rectify it please.

Responses 2 and 3:

First, we apologize if there was any confusion on the categorization of education levels. Education was indeed ranked from lowest to highest in both Tables 1 and 2: High school, CEGEP/Trade certificate/College and University. CEGEP in Québec is roughly the equivalent of College in the Anglo-Saxon system. To facilitate comprehension, the “or less” statement was modified to “or no diploma” and we have removed the “or more” associated with the University degree throughout the revised manuscript (Lines 96, 105-106, 135-137, 186-188, 191; Tables 1 and 2).  

We have also toned down the interpretation of the unexpected association between SFA intake and education levels in the discussion (lines 185-194).

Point 4:

Perhaps here other factors such as sedentarism, practice or not of healthy habits, sugar consumption, etc. my be affecting the data. Particularly in relation to just a 0,5% higher SFA intake for the obese individuals in the study. I would expect increases higher than 1-2% to be the cause to become obese.  Can the authors develop/discuss more about this little increase in SFA intake for the obese in relation to other possible factors.

Response 4:

Thank you for this comment. We have already mentioned the possibility that such results are affected by known under-reporting in CCHS 2015 (Garriguet, 2018), particularly among obese individuals who have been shown to be more susceptible to under-reporting (Murakami et al, 2015; Jessri et al, 2016). Nonetheless, weight status can also be influenced by many factors such as diet, sedentarism and the practice or not of other lifestyle factors (smoking, drinking, stress, etc.), which were not considered in our analyses. Therefore, the possibility that such factors may have contributed to attenuating or emphasizing the association between SFA intake and obesity cannot be ruled out. This possibility is briefly discussed in the revised manuscript (Lines 202-204).